# Learning Efficient Random Maximum A-Posteriori Predictors with Non-Decomposable Loss Functions

**Tamir Hazan**
University of Haifa

**Subhransu Maji**
TTI Chicago

**Joseph Keshet**
Bar-Ilan university

**Tommi Jaakkola**
CSAIL, MIT

## Abstract

In this work we develop efficient methods for learning random MAP predictors for structured label problems. In particular, we construct posterior distributions over perturbations that can be adjusted via stochastic gradient methods. We show that any smooth posterior distribution would suffice to define a smooth PAC-Bayesian risk bound suitable for gradient methods. In addition, we relate the posterior distributions to computational properties of the MAP predictors. We suggest multiplicative posteriors to learn super-modular potential functions that accompany specialized MAP predictors such as graph-cuts. We also describe label-augmented posterior models that can use efficient MAP approximations, such as those arising from linear program relaxations.

## 1 Introduction

Learning and inference in complex models drives much of the research in machine learning applications ranging from computer vision, natural language processing, to computational biology [1, 18, 21]. The inference problem in such cases involves assessing the likelihood of possible structured-labels, whether they be objects, parsers, or molecular structures. Given a training dataset of instances and labels, the learning problem amounts to estimation of the parameters of the inference engine, so as to best describe the labels of observed instances. The goodness of fit is usually measured by a loss function.

The structures of labels are specified by assignments of random variables, and the likelihood of the assignments are described by a potential function. Usually, it is feasible to only find the most likely or maximum a-posteriori (MAP) assignment, rather than sampling according to their likelihood. Indeed, substantial effort has gone into developing algorithms for recovering MAP assignments, either based on specific parametrized restrictions such as super-modularity [2] or by devising approximate methods based on linear programming relaxations [21]. Learning MAP predictors is usually done by structured-SVMs that compare a "loss adjusted" MAP prediction to its training label [25]. In practice, most loss functions used decompose in the same way as the potential function, so as to not increase the complexity of the MAP prediction task. Nevertheless, non-decomposable loss functions capture the structures in the data that we would like to learn.

Bayesian approaches for expected loss minimization, or risk, effortlessly deal with non-decomposable loss functions. The inference procedure samples a structure according to its likelihood, and computes its loss given a training label. Recently [17, 23] constructed probability models through MAP predictions. These "perturb-max" models describe the robustness of the MAP prediction to random changes of its parameters. Therefore, one can draw unbiased samples from these distributions using MAP predictions. Interestingly, when incorporating perturb-max models to Bayesian loss minimization one would ultimately like to use the PAC-Bayesian risk [11, 19, 3, 20, 5, 10].

Our work explores the Bayesian aspects that emerge from PAC-Bayesian risk minimization. We focus on computational aspects when constructing posterior distributions, so that they could be used

to minimize the risk bound efficiently. We show that any smooth posterior distribution would suffice to define a smooth risk bound which can be minimized through gradient decent. In addition, we relate the posterior distributions to the computational properties of MAP predictors. We suggest multiplicative posterior models to learn super-modular potential functions, that come with specialized MAP predictors such as graph-cuts [2]. We also describe label-augmented posterior models that can use MAP approximations, such as those arising from linear program relaxations [21].

## 2  Background

Learning complex models typically involves reasoning about the states of discrete variables whose labels (assignments of values) specify the discrete structures of interest. The learning task which we consider in this work is to fit parameters $w$ that produce to most accurate prediction $y \in Y$ to a given object $x$. Structures of labels are conveniently described by a discrete product space $Y = Y_1 \times \cdots \times Y_n$. We describe the potential of relating a label $y$ to an object $x$ with respect to the parameters $w$ by real valued functions $\theta(y; x, w)$. Our goal is to learn the parameters $w$ that best describe the training data $(x, y) \in S$. Within Bayesian perspectives, the distribution that one learns given the training data is composed from a distribution over the parameter space $q_w(\gamma)$ and over the labels space $P[y|w, x] \propto \exp \theta(y; x, w)$. Using the Bayes rule we derive the predictive distribution over the structures

$$P[y|x] = \int P[y|\gamma, x] q_w(\gamma) d\gamma \tag{1}$$

Unfortunately, sampling algorithms over complex models are provably hard in theory and tend to be slow in many cases of practical interest [7]. This is in contrast to the maximum a-posteriori (MAP) prediction, which can be computed efficiently for many practical cases, even when sampling is provably hard.

$$\text{(MAP predictor)} \qquad y_w(x) = \arg \max_{y_1, \ldots, y_n} \theta(y; x, w) \tag{2}$$

Recently, [17, 23] suggested to change of the Bayesian posterior probability models to utilize the MAP prediction in a deterministic manner. These perturb-max models allow to sample from the predictive distribution with a single MAP prediction:

$$\text{(Perturb-max models)} \qquad P[y|x] \stackrel{def}{=} P_{\gamma \sim q_w} \big[ y = y_\gamma(x) \big] \tag{3}$$

A potential function is decomposed along a graphical model if it has the form $\theta(y; x, w) = \sum_{i \in V} \theta_i(y_i; x, w) + \sum_{i,j \in E} \theta_{i,j}(y_i, y_j; x, w)$. If the graph has no cycles, MAP prediction can be computed efficiently using the belief propagation algorithm. Nevertheless, there are cases where MAP prediction can be computed efficiently for graph with cycles. A potential function is called supermodular if it is defined over $Y = \{-1, 1\}^n$ and its pairwise interactions favor adjacent states to have the same label, i.e., $\theta_{i,j}(-1, -1; x, w) + \theta_{i,j}(1, 1; x, w) \geq \theta_{i,j}(-1, 1; x, w) + \theta_{i,j}(1, -1; x, w)$. In such cases MAP prediction reduces to computing the min-cut (graph-cuts) algorithm.

Recently, a sequence of works attempt to solve the MAP prediction task for non-supermodular potential function as well as general regions. These cases usually involve potentials function that are described by a family $R$ of subsets of variables $r \subset \{1, \ldots, n\}$, called regions. We denote by $y_r$ the set of labels that correspond to the region $r$, namely $(y_i)_{i \in r}$ and consider the following potential functions $\theta(y; x, w) = \sum_{r \in R} \theta_r(y_r; x, w)$. Thus, MAP prediction can be formulated as an integer linear program:

$$b^* \in \arg \max_{b_r(y_r)} \sum_{r, y_r} b_r(y_r) \theta_r(y_r; x, w) \tag{4}$$

$$s.t. \qquad b_r(y_r) \in \{0, 1\}, \sum_{y_r} b_r(y_r) = 1, \quad \sum_{y_s \backslash y_r} b_s(y_s) = b_r(y_r) \quad \forall r \subset s$$

The correspondence between MAP prediction and integer linear program solutions is $(y_w(x))_i = \arg \max_{y_i} b_i^*(y_i)$. Although integer linear program solvers provide an alternative to MAP prediction, they may be restricted to problems of small size. This restriction can be relaxed when one replaces the integral constraints $b_r(y_r) \in \{0, 1\}$ with nonnegative constraints $b_r(y_r) \geq 0$. These

linear program relaxations can be solved efficiently using different convex max-product solvers, and whenever these solvers produce an integral solution it is guaranteed to be the MAP prediction [21].

Given training data of object-label pairs, the learning objective is to estimate a predictive distribution over the structured-labels. The goodness of fit is measured by a loss function $L(\hat{y}, y)$. As we focus on randomized MAP predictors our goal is to learn the parameters $w$ that minimize the expected perturb-max prediction loss, or randomized risk. We define the randomized risk at a single instance-label pair as

$$R(w, x, y) = \sum_{\hat{y} \in Y} P_{\gamma \sim q_w}\big[\hat{y} = y_\gamma(x)\big] L(\hat{y}, y).$$

Alternatively, the randomized risk takes the form $R(w, x, y) = E_{\gamma \sim q_w}[L(y_\gamma(x), y)]$. The randomized risk originates within the PAC-Bayesian generalization bounds. Intuitively, if the training set is an independent sample, one would expect that best predictor on the training set to perform well on unlabeled objects at test time.

## 3 Minimizing PAC-Bayesian generalization bounds

Our approach is based on the PAC-Bayesian risk analysis of random MAP predictors. In the following we state the PAC-Bayesian generalization bound for structured predictors and describe the gradients of these bounds for any smooth posterior distribution.

The PAC-Bayesian generalization bound describes the expected loss, or randomized risk, when considering the true distributions over object-labels in the world $R(w) = E_{(x,y) \sim \rho}[R(w, x, y)]$. It upper bounds the randomized risk by the empirical randomized risk $R_S(w) = \frac{1}{|S|} \sum_{(x,y) \in S} R(w, x, y)$ and a penalty term which decreases proportionally to the training set size. Here we state the PAC-Bayesian theorem, that holds uniformly for all posterior distributions over the predictions.

**Theorem 1.** *(Catoni [3], see also [5]). Let $L(\hat{y}, y) \in [0, 1]$ be a bounded loss function. Let $p(\gamma)$ be any probability density function and let $q_w(\gamma)$ be a family of probability density functions parameterized by $w$. Let $KL(q_w||p) = \int q_w(\gamma) \log(q_w(\gamma)/p(\gamma))$. Then, for any $\delta \in (0, 1]$ and for any real number $\lambda > 0$, with probability at least $1 - \delta$ over the draw of the training set the following holds simultaneously for all $w$*

$$R(w) \leq \frac{1}{1 - \exp(-\lambda)} \left( \lambda R_S(w) + \frac{KL(q_w||p) + \log(1/\delta)}{|S|} \right)$$

For completeness we present a proof sketch for the theorem in the appendix. This proof follows Seeger's PAC-Bayesian approach [19], and extended to the structured label case [13]. The proof technique replaces prior randomized risk, with the posterior randomized risk that holds uniformly for every $w$, while penalizing this change by their KL-divergence. This change-of-measure step is close in spirit to the one that is performed in importance sampling. The proof is then concluded by simple convex bound on the moment generating function of the empirical risk.

To find the best posterior distribution that minimizes the randomized risk, one can minimize its empirical upper bound. We show that whenever the posterior distributions have smooth probability density functions $q_w(\gamma)$, the perturb-max probability model is smooth as a function of $w$. Thus the randomized risk bound can be minimized with gradient methods.

**Theorem 2.** *Assume $q_w(\gamma)$ is smooth as a function of its parameters, then the PAC-Bayesian bound is smooth as a function of $w$:*

$$\nabla_w R_S(w) = \frac{1}{|S|} \sum_{(x,y) \in S} E_{\gamma \sim q_w} \Big[ \nabla_w[\log q_w(\gamma)] L(y_\gamma(x), y) \Big]$$

*Moreover, the KL-divergence is a smooth function of $w$ and its gradient takes the form:*

$$\nabla_w KL(q_w||p) = E_{\gamma \sim q_w} \Big[ \nabla_w[\log q_w(\gamma)] \big( \log(q_w(\gamma)/p(\gamma)) + 1 \big) \Big]$$

**Proof:** First we note that $R(w, x, y) = \int q_w(\gamma) L(y_\gamma(x), y) d\gamma$. Since $q_w(\gamma)$ is a probability density function and $L(\hat{y}, y) \in [0, 1]$ we can differentiate under the integral (cf. [4] Theorem 2.27).

$$\nabla_w R(w, x, y) = \int \nabla_w q_w(\gamma) L(y_\gamma(x), y) d\gamma$$

Using the identity $\nabla_w q_w(\gamma) = q_w(\gamma)\nabla_w \log(q_w(\gamma))$ the first part of the proof follows. The second part of the proof follows in the same manner, while noting that $\nabla_w(q_w(\gamma)\log q_w(\gamma)) = (\nabla_w q_w(\gamma))(\log q_w(\gamma) + 1)$. $\square$

The gradient of the randomized empirical risk is governed by the gradient of the log-probability density function of its corresponding posterior model. For example, Gaussian model with mean $w$ and identity covariance matrix has the probability density function $q_w(\gamma) \propto \exp(-\|\gamma - w\|^2/2)$, thus the gradient of its log-density is the linear moment $\gamma$, i.e., $\nabla_w[\log q_w] = \gamma - w$.

Taking any smooth distribution $q_w(\gamma)$, we can find the parameters $w$ by descending along the stochastic gradient of the PAC-Bayesian generalization bound. The gradient of the randomized empirical risk is formed by two expectations, over the sample points and over the posterior distribution. Computing these expectations is time consuming, thus we use a single sample $\nabla_\gamma[\log q_w(\gamma)]L(y_\gamma(x), y)$ as an unbiased estimator for the gradient. Similarly we estimate the gradient of the KL-divergence with an unbiased estimator which requires a single sample of $\nabla_w[\log q_w(\gamma)](\log(q_w(\gamma)/p(\gamma)) + 1)$. This approach, called stochastic approximation or online gradient descent, amounts to use the stochastic gradient update rule

$$ w \leftarrow w - \eta \cdot \lambda \nabla_w[\log q_w(\gamma)]\Big(L(y_\gamma(x), y) + \log(q_w(\gamma)/p(\gamma)) + 1\Big) $$

where $\eta$ is the learning rate. Next, we explore different posterior distributions from computational perspectives. Specifically, we show how to learn the posterior model so to ensure the computational efficiency of its MAP predictor.

## 4 Learning posterior distributions efficiently

The ability to efficiently apply MAP predictors is key to the success of the learning process. Although MAP predictions are NP-hard in general, there are posterior models for which they can be computed efficiently. For example, whenever the potential function corresponds to a graphical model with no cycles, MAP prediction can be efficiently computed for any learned parameters $w$.

Learning unconstrained parameters with random MAP predictors provides some freedom in choosing the posterior distribution. In fact, Theorem 2 suggests that one can learn any posterior distribution by performing gradient descent on its risk bound, as long as its probability density function is smooth. We show that for unconstrained parameters, additive posterior distributions simplify the learning problem, and the complexity of the bound (i.e., its KL-divergence) mostly depends on its prior distribution.

**Corollary 1.** *Let $q_0(\gamma)$ be a smooth probability density function with zero mean and set the posterior distribution using additive shifts $q_w(\gamma) = q_0(\gamma - w)$. Let $H(q) = -E_{\gamma \sim q}[\log q(\gamma)]$ be the entropy function. Then*

$$ KL(q_w||p) = -H(q_0) - E_{\gamma \sim q_0}[\log p(\gamma + w)] $$

*In particular, if $p(\gamma) \propto \exp(-\|\gamma\|^2)$ is Gaussian then $\nabla_w KL(q_w||p) = w$*

**Proof:** $KL(q_w||p) = -H(q_w) - E_{\gamma \sim q_w}[\log p(\gamma)]$. By a linear change of variable, $\hat\gamma = \gamma - w$ it follows that $H(q_w) = H(q_0)$ thus $\nabla_w H(q_w) = 0$. Similarly $E_{\gamma \sim q_w}[\log p(\gamma)] = E_{\gamma \sim q_0}[\log p(\gamma + w)]$. Finally, if $p(\gamma)$ is Gaussian then $E_{\gamma \sim q_0}[\log p(\gamma + w)] = -w^2 - E_{\gamma \sim q_0}[\gamma^2]$. $\square$

This result implies that every additively-shifted smooth posterior distribution may consider the KL-divergence penalty as the square regularization when using a Gaussian prior $p(\gamma) \propto \exp(-\|\gamma\|^2)$. This generalizes the standard claim on Gaussian posterior distributions [11], for which $q_0(\gamma)$ are Gaussians. Thus one can use different posterior distributions to better fit the randomized empirical risk, without increasing the computational complexity over Gaussian processes.

Learning unconstrained parameters can be efficiently applied to tree structured graphical models. This, however, is restrictive. Many practical problems require more complex models, with many cycles. For some of these models linear program solvers give efficient, although sometimes approximate, MAP predictions. For supermodular models there are specific solvers, such as graph-cuts, that produce fast and accurate MAP predictions. In the following we show how to define posterior distributions that guarantee efficient predictions, thus allowing efficient sampling and learning.

### 4.1 Learning constrained posterior models

MAP predictions can be computed efficiently in important practical cases, e.g., supermodular potential functions satisfying $\theta_{i,j}(-1,-1;x,w) + \theta_{i,j}(1,1;x,w) \geq \theta_{i,j}(-1,1;x,w) + \theta_{i,j}(1,-1;x,w)$. Whenever we restrict ourselves to symmetric potential function $\theta_{i,j}(y_i, y_j; x, w) = w_{i,j} y_i y_j$, supermodularity translates to nonnegative constraint on the parameters $w_{i,j} \geq 0$. In order to model posterior distributions that allow efficient sampling we define models over the constrained parameter space. Unfortunately, the additive posterior models $q_w(\gamma) = q_0(\gamma - w)$ are inappropriate for this purpose, as they have a positive probability for negative $\gamma$ values and would generate non-supermodular models.

To learn constrained parameters one requires posterior distributions that respect these constraints. For nonnegative parameters we apply posterior distributions that are defined on the nonnegative real numbers. We suggest to incorporate the parameters of the posterior distribution in a multiplicative manner into a distribution over the nonnegative real numbers. For any distribution $q_\alpha(\gamma)$ we determine a posterior distribution with parameters $w$ as $q_w(\gamma) = q_\alpha(\gamma/w)/w$. We show that multiplicative posterior models naturally provide log-barrier functions over the constrained set of nonnegative numbers. This property is important to the computational efficiency of the bound minimization algorithm.

**Corollary 2.** *For any probability distribution $q_\alpha(\gamma)$, let $q_{\alpha,w}(\gamma) = q_\alpha(\gamma/w)/w$ be the parametrized posterior distribution. Then*

$$KL(q_{\alpha,w}||p) = -H(q_\alpha) - \log w - E_{\gamma \sim q_\alpha}[\log p(w\gamma)]$$

*Define the Gamma function $\Gamma(\alpha) = \int_0^\infty \gamma^{\alpha-1} \exp(-\gamma)$. If $p(\gamma) = q_\alpha(\gamma) = \gamma^{\alpha-1} \exp(-\gamma)/\Gamma(\alpha)$ have the Gamma distribution with parameter $\alpha$, then $E_{\gamma \sim q_\alpha}[\log p(w\gamma)] = (\alpha-1)\log w - \alpha w$. Alternatively, if $p(\gamma)$ are truncated Gaussians then $E_{\gamma \sim q_\alpha}[\log p(w\gamma)] = -\frac{\alpha}{2}w^2 + \log\sqrt{\pi/2}$.*

**Proof:** The entropy of multiplicative posterior models naturally implies the log-barrier function:

$$-H(q_{\alpha,w}) \overset{\hat{\gamma}=\gamma/w}{=} \int q_\alpha(\hat{\gamma})\Big(\log q_\alpha(\hat{\gamma}) - \log w\Big) d\hat{\gamma} = -H(q_\alpha) - \log w.$$

Similarly, $E_{\gamma \sim q_{\alpha,w}}[\log p(\gamma)] = E_{\gamma \sim q_\alpha}[\log p(w\gamma)]$. The special cases for the Gamma and the truncated normal distribution follow by a direct computation. $\square$

The multiplicative posterior distribution would provide the barrier function $-\log w$ as part of its KL-divergence. Thus the multiplicative posterior effortlessly enforces the constraints of its parameters. This property suggests that using multiplicative rules are computationally favorable. Interestingly, using a prior model with Gamma distribution adds to the barrier function a linear regularization term $\|w\|_1$ that encourages sparsity. On the other hand, a prior model with a truncated Gaussian adds a square regularization term which drifts the nonnegative parameters away from zero. A computational disadvantage of the Gaussian prior is that its barrier function cannot be controlled by a parameter $\alpha$.

### 4.2 Learning posterior models with approximate MAP predictions

MAP prediction can be phrased as an integer linear program, stated in Equation (4). The computational burden of integer linear programs can be relaxed when one replaces the integral constraints with nonnegative constraints. This approach produces approximate MAP predictions. An important learning challenge is to extend the predictive distribution of perturb-max models to incorporate approximate MAP solutions. Approximate MAP predictions are are described by the feasible set of their linear program relaxations, that is usually called the local polytope:

$$L(R) = \Big\{ b_r(y_r) : \; b_r(y_r) \geq 0, \; \sum_{y_r} b_r(y_r) = 1, \; \forall r \subset s \; \sum_{y_s \setminus y_r} b_s(y_s) = b_r(y_r) \Big\}$$

Linear programs solutions are usually the extreme points of their feasible polytope. The local polytope is defined by a finite set of equalities and inequalities, thus it has a finite number of extreme points. The perturb-max model that is defined in Equation (3) can be effortlessly extended to the finite set of the local polytope extreme points [15]. This approach has two flaws. First, linear program solutions might not be extreme points, and decoding such a point usually requires additional

computational effort. Second, without describing the linear program solutions one cannot incorporate loss functions that take the structural properties of approximate MAP predictions into account when computing the the randomized risk.

**Theorem 3.** *Consider approximate MAP predictions that arise from relaxation of the MAP prediction problem in Equation (4).*

$$\arg\max_{b_r(y_r)} \sum_{r,y_r} b_r(y_r)\theta_r(y_r; x, w) \quad s.t. \quad b \in L(R)$$

*Then any optimal solution $b^*$ is described by a vector $\tilde{y}_w(x)$ in the finite power sets over the regions, $\tilde{Y} \subset \times_r 2^{Y_r}$:*

$$\tilde{y}_w(x) = (\tilde{y}_{w,r}(x))_{r \in \mathcal{R}} \qquad where \qquad \tilde{y}_{w,r}(x) = \{y_r : b_r^*(y_r) > 0\}$$

*Moreover, if there is a unique optimal solution $b^*$ then it corresponds to an extreme point in the local polytope.*

**Proof:** The program is convex over a compact set, thus strong duality holds. Fixing the Lagrange multipliers $\lambda_{r \to s}(y_r)$ that correspond to the marginal constraints $\sum_{y_s \setminus y_r} b_s(y_s) = b_r(y_r)$, and considering the probability constraints as the domain of the primal program, we derive the dual program

$$\sum_r \max_{y_r} \left\{ \theta_r(y_r; x, w) + \sum_{c:c \subset r} \lambda_{c \to r}(y_c) - \sum_{p:p \supset r} \lambda_{r \to p}(y_r) \right\}$$

Lagrange optimality constraints (or equivalently, Danskin Theorem) determine the primal optimal solutions $b_r^*(y_r)$ to be probability distributions over the set $\arg\max_{y_r}\{\theta_r(y_r; x, w) + \sum_{c:c \subset r} \lambda_{c \to r}^*(y_c) - \sum_{p:p \supset r} \lambda_{r \to p}^*(y_r)\}$ that satisfy the marginalization constraints. Thus $\tilde{y}_{w,r}(x)$ is the information that identifies the primal optimal solutions, i.e., any other primal feasible solution that has the same $\tilde{y}_{w,r}(x)$ is also a primal optimal solution. □

This theorem extends Proposition 3 in [6] to non-binary and non-pairwise graphical models. The theorem describes the discrete structures of approximate MAP predictions. Thus we are able to define posterior distributions that use efficient, although approximate, predictions while taking into account their structures. To integrate these posterior distributions to randomized risk we extend the loss function to $L(\tilde{y}_w(x), y)$. One can verify that the results in Section 3 follow through, e.g., by considering loss functions $L : \tilde{Y} \times \tilde{Y} \to [0, 1]$ while the training examples labels belong to the subset $Y \subset \tilde{Y}$.

## 5 Empirical evaluation

We perform experiments on an interactive image segmentation. We use the Grabcut dataset proposed by Blake et al. [1] which consists of 50 images of objects on cluttered backgrounds and the goal is to obtain the pixel accurate segmentations of the object given an initial "trimap" (see Figure 1). A trimap is an approximate segmentation of the image into regions that are well inside, well outside and the boundary of the object, something a user can easily specify in an interactive application.

A popular approach for segmentation is the GrabCut approach [2, 1]. We learn parameters for the "Gaussian Mixture Markov Random Field" (GMMRF) formulation of [1] using a potential function over foreground/background segmentations $Y = \{-1, 1\}^n$: $\theta(y; x, w) = \sum_{l \in V} \theta_i(y_i; x, w) + \sum_{i,j \in E} \theta_{i,j}(y_i, y_j; x, w)$. The local potentials are $\theta_i(y_i; x, w) = w_{y_i} \log P(y_i|x)$, where $w_{y_i}$ are parameters to be learned while $P(y_i|x)$ are obtained from a Gaussian mixture model learned on the background and foreground pixels for an image $x$ in the initial trimap. The pairwise potentials are $\theta_{i,j}(y_i, y_j; x, w) = w_a \exp(-(x_i - x_j)^2)y_iy_j$, where $x_i$ denotes the intensity of image $x$ at pixel $i$, and $w_a$ are the parameters to be learned for the angles $a \in \{0, 90, 45, -45\}^\circ$. These potential functions are supermodular as long as the parameters $w_a$ are nonnegative, thus MAP prediction can be computed efficiently with the graph-cuts algorithm. For these parameters we use multiplicative posterior model with the Gamma distribution. The dataset does not come with a standard training/test split so we use the odd set of images for training and even set of images for testing. We use stochastic gradient descent with the step parameter decaying as $\eta_t = \frac{\eta}{t_o+t}$ for 250 iterations.

| Method | Grabcut loss | PASCAL loss |
|---|---|---|
| Our method | **7.77%** | **5.29%** |
| Structured SVM (hamming loss) | 9.74% | 6.66% |
| Structured SVM (all-zero loss) | 7.87% | 5.63% |
| GMMRF (Blake et al. [1]) | 7.88% | 5.85% |
| Perturb-and-MAP ([17]) | 8.19% | 5.76% |

Table 1: Learning the Grabcut segmentations using two different loss functions. Our learned parameters outperform structured SVM approaches and Perturb-and-MAP moment matching

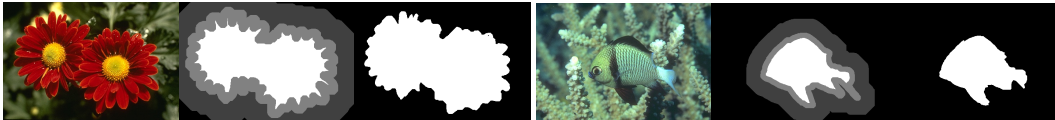

Figure 1: Two examples of image (*left*), input "trimap" (*middle*) and the final segmentation (*right*) produced using our learned parameters.

We use two different loss functions for training/testing for our approach to illustrate the flexibility of our approach for learning using various task specific loss functions. The "GrabCut loss" measures the fraction of incorrect pixels labels in the region specified as the boundary in the trimap. The "PASCAL loss", which is commonly used in several image segmentation benchmarks, measures the ratio of the intersection and union of the foregrounds of ground truth segmentation and the solution.

As a comparison we also trained parameters using moment matching of MAP perturbations [17] and structured SVM. We use a stochastic gradient approach with a decaying step size for 1000 iterations. Using structured SVM, solving loss-augmented inference $\max_{\hat{y} \in Y}\{L(y, \hat{y}) + \theta(y; x, w)\}$ with the hamming loss can be efficiently done using graph-cuts. We also consider learning parameters with all-zero loss function, i.e., $L(y, \hat{y}) \equiv 0$. To ensure that the weights remain non-negative we project the weights into the non-negative side after each iteration.

Table 1 shows the results of learning using various methods. For the GrabCut loss, our method obtains comparable results to the GMMRF framework of [1], which used hand-tuned parameters. Our results are significantly better when PASCAL loss is used. Our method also outperforms the parameters learned using structured SVM and Perturb-and-MAP approaches. In our experiments the structured SVM with the hamming loss did not perform well – the loss augmented inference tended to focus on maximum violations instead of good solutions which causes the parameters to change even though the MAP solution has a low loss (a similar phenomenon was observed in [22]). Using the all-zero loss tends to produce better results in practice as seen in Table 1. Figure 1 shows some examples images, the input trimap, and the segmentations obtained using our approach.

## 6 Related work

Recent years have introduced many optimization techniques that lend efficient MAP predictors for complex models. These MAP predictors can be integrated to learn complex models using structured-SVM [25]. Structured-SVM has a drawback, as its MAP prediction is adjusted by the loss function, therefore it has an augmented complexity. Recently, there has been an effort to efficiently integrate non-decomposable loss function into structured-SVMs [24]. However this approach does not hold for any loss function.

Bayesian approaches to loss minimization treat separately the prediction process and the loss incurred, [12]. However, the Bayesian approach depends on the efficiency of its sampling procedure, but unfortunately, sampling in complex models is harder that the MAP prediction task [7].

The recent works [17, 23, 8, 9, 16] integrate efficient MAP predictors into Bayesian modeling. [23] describes the Bayesian perspectives, while [17, 8] describe their relations to the Gibbs distribution and moment matching. [9] provide unbiased samples form the Gibbs distribution using MAP predictors and [16] present their measure concentration properties. Other strategies for producing

(pseudo) samples efficiently include Herding [26]. However, these approaches do not consider risk minimization.

The perturb-max models in Equation (3) play a key role in PAC-Bayesian theory [14, 11, 19, 3, 20, 5, 10]. The PAC-Bayesian approaches focus on generalization bounds to the object-label distribution. However, the posterior models in the PAC-Bayesian approaches were not extensively studied in the past. In most cases the posterior model remained undefined. [10] investigate linear predictors with Gaussian posterior models to have a structured-SVM like bound. This bound holds uniformly for every $\lambda$ and its derivation is quite involved. In contrast we use Catoni's PAC-Bayesian bound that is not uniform over $\lambda$ but does not require the $\log |S|$ term [3, 5]. The simplicity of Catoni's bound (see Appendix) makes it amenable to different extensions. In our work, we extend these results to smooth posterior distributions, while maintaining the quadratic regularization form. We also describe posterior distributions for non-linear models. In different perspective, [3, 5] describe the optimal posterior, but unfortunately there is no efficient sampling procedure for this posterior model. In contrast, our work explores posterior models which allow efficient sampling. We investigate two posterior models: the multiplicative models, for constrained MAP solvers such as graph-cuts, and posterior models for approximate MAP solutions.

## 7 Discussion

Learning complex models requires one to consider non-decomposable loss functions that take into account the desirable structures. We suggest to use the Bayesian perspectives to efficiently sample and learn such models using random MAP predictions. We show that any smooth posterior distribution would suffice to define a smooth PAC-Bayesian risk bound which can be minimized using gradient decent. In addition, we relate the posterior distributions to the computational properties of the MAP predictors. We suggest multiplicative posterior models to learn supermodular potential functions that come with specialized MAP predictors such as graph-cuts algorithm. We also describe label-augmented posterior models that can use efficient MAP approximations, such as those arising from linear program relaxations. We did not evaluate the performance of these posterior models and further explorations of such models is required.

The results here focus on posterior models that would allow for efficient sampling using MAP predictions. There are other cases for which specific posterior distributions might be handy, e.g., learning posterior distributions of Gaussian mixture models. In these cases, the parameters include the covariance matrix, thus would require to sample over the family of positive definite matrices.

## A   Proof sketch for Theorem 1

Theorem 2.1 in [5]: For any distribution $D$ over object-labels pairs, for any $w$-parametrized distribution $q_w$, for any prior distribution $p$, for any $\delta \in (0, 1]$, and for any convex function $\mathcal{D} : [0, 1] \times [0, 1] \rightarrow R$, with probability at least $1 - \delta$ over the draw of the training set the divergence $\mathcal{D}(E_{\gamma \sim q_w} R_S(\gamma), E_{\gamma \sim q_w} R(\gamma))$ is upper bounded simultaneously for all $w$ by

$$\frac{1}{|S|} \Big[ KL(q_w || p) + \log \Big( \frac{1}{\delta} E_{\gamma \sim p} E_{S \sim D^m} \exp \big( m \mathcal{D}(R_S(\gamma), R(\gamma)) \big) \Big) \Big]$$

For $\mathcal{D}(R_S(\gamma), R(\gamma)) = \mathcal{F}(R(\gamma)) - \lambda R_S(\gamma)$, the bound reduces to a simple convex bound on the moment generating function of the empirical risk: $E_{S \sim D^m} \exp \big( m \mathcal{D}(R_S(\gamma, x, y), R(\gamma, x, y)) \big) = \exp(m\mathcal{F}(R(\gamma))) E_{S \sim D^m} \exp(-m\lambda R_S(\gamma))$ Since the exponent function is a convex function of $R_S(\gamma) = R_S(\gamma) \cdot 1 + (1 - R_S(\gamma)) \cdot 0$, the moment generating function bound is $\exp(-\lambda R_S(\gamma)) \leq R_S(\gamma) \exp(-\lambda) + (1 - R_S(\gamma))$. Since $E_S R_S(\gamma) = R(\gamma)$, the right term in the risk bound in can be made 1 when choosing $\mathcal{F}(R(\gamma))$ to be the inverse of the moment generating function bound. This is Catoni's bound [3, 5] for the structured labels case. To derive Theorem 1 we apply $\exp(-x) \leq 1 - x$ to derive the lower bound $(1 - \exp(-\lambda)) E_{\gamma \sim q_w} R(\gamma) - \lambda E_{\gamma \sim q_w} R_S(\gamma) \leq \mathcal{D}(E_{\gamma \sim q_w} R_S(\gamma), E_{\gamma \sim q_w} R(\gamma))$.

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
