[Reviews · NeurIPS 2013]

Submitted by Assigned_Reviewer_6

The paper adapts a random MAP predictor model for structured prediction, and proposes to learn the model by minimizing the PAC-bayesian bound. They discussed different possible choices of the posterior distributions, including multiplicative models of non-negative parameters for sub-modular models, and posterior distributions for approximate MAP solutions (but without much details) .

Quality: median.

1. The paper compares with structured SVM. But more natural comparisons may be methods in [6,22] which also learns random MAP predictions. It would be nice to provide experimental comparison to these methods.

2. It would be nice to present results on more than one datasets.

Clarity. The paper is well written overall. But some part of Section 4 confuses me a bit. In particular,

1). Corollary 1 assumes additive posterior distributions, but right after it, the statement that "every smooth posterior distribution remains quadratic form" seems to over-generalize the result to any smooth posterior distribution. Since this is considered as one of the important contribution to the paper. Please you explain this point more clearly and explicitly in the paper.

2). Section 4.1 proposes to build posterior distributions directly for approximate MAP predictions. The idea is interesting, but it seems that no solid example of this kind is outlined, and that is no related experiments provided (am i correct?). This makes me feel that this section is more a "future-looking" discussion. If this true, please state this fact explicitly in the paper, since otherwise it may confuse the readers. Also, is it necessary to state Theorem 3 as a theorem? This is really a quite obvious result; stating it as a theorem may make readers feel that this is an important important contribution of the paper (especially in contrast to the other two corollaries you used). Give that no solid result is provided in section 4.1, I suggest to significantly shorten this section and possibly rearrange its position (e.g., putting it after Section 4.2).

Originality: median or low. The paper learns a random MAP predictor model for structured prediction by directly minimizing the PAC-bayesian bound. The random MAP framework has been proposed in [6,22, 8] and the loss minimization method is basically the same as the method in [8] (probably the main difference is that [8] restricts on additive and Gaussian posterior, but the technical difference seems very minor). I think the real contribution of this paper is on proposing to use more general posterior distributions, in particular, a multiplicative models of non-negative parameters for sub-modular models, and (potentially) posterior distributions for approximate MAP solutions. But note that the paper only shows results on multiplicative posterior models; although section 4.1 talks about building posterior models for approximate MAP, the paper does not show any concrete example of this kind.

Significance. The method proposed in the paper may be useful to improve accuracy of structured prediction problems.

Summary: The paper discusses a class of interesting algorithm for structured prediction. But the originality of the paper seems limited. The clarity should be improved.

Submitted by Assigned_Reviewer_7

This paper addresses learning a structured prediction model under the PAC-bayesian framework. In particular, it considers the case where the loss function is non-decomposable (therefore loss-augmented inference in large margin learning is difficult) and the model is amenable to MAP queries.
Let q be the posterior over model parameters, the proposed learning algorithm (one gradient step) involves taking the gradient of log q, sampling from q, MAP query from the model, and computing the loss function. So the method is applicable for any choice of q and model such that the above operations can be carried out efficiently.
The authors then discussed two special cases: (1) when LP relaxation is used in place of the MAP query, the proposed framework can be applied by extending the loss function to non-integer vertices of the local polytope; (2) constraints on model parameters can be imposed to ensure the applicability of certain MAP inference methods (such as graphcut)
Only (2) is demonstrated in experiments..

From a structured prediction practitioner’s point of view, the proposed method appears to be an interesting alternative to margin based approaches that require loss-augmented inference, which is hard with arbitrary loss functions. And using the “right” loss function has been demonstrated to be important by other researchers.

Minor point:
In theorem 1, the symbol ‘p’ (I assume it’s the prior) is not defined.

I have read the authors' response and other reviews.

Summary: The proposed method appears to be interesting and useful in a practitioner's point of view.

Submitted by Assigned_Reviewer_8

This paper discusses learning models based on random Maximum a
Posteriori (random MAP) perturbations so as to directly optimize
non-decomposable loss functions.

The main contributions are as follows:

(1) framing the learning problem as minimizing a PAC-Bayes
generalization bound

(2) observing that so long as the perturbation distribution is smooth,
the expected loss component of the PAC-Bayes bound is smooth.

(3) extending the method to work when MAP solutions are solutions to a
linear program relaxation

(4) enforcing positivity constraints on parameters (e.g. to guarantee
submodularity for the MAP problem) by modeling certain parameters
using distributions that only give support to positive values

Experiments are reported on a GrabCut-style segmentation task, and
results show modest gains over structural SVM.


Strengths:

* Previous work on perturbation models hasn't considered direct
optimization of expected loss, and the proposal in this paper is the
natural way to do it.

* The framing in terms of PAC Bayes is nice and gives some theoretical
justification for the strategy.

* Dealing with approximate inference and positivity constraints are
important practical concerns, and the methods presented for doing so
are natural.


Weaknesses:

* The PAC-Bayes framing seems very similar to that of Keshet et al
(cited as [8]). I think a discussion of the similarities/differences
is warranted.

* I'm concerned about the variance of the gradient discussed in line
178. It is essentially relying upon random parameter perturbations to
find good directions in which to move the parameters. I'd like to see
experiments or analysis explaining what happens as the dimension of w
grows, and also how the gradient behaves with a variety of loss
functions.

* Relatedly, the experiments don't help in revealing interesting
things about what is happening in the learning. We just have a small
table of numbers showing that the proposed method is best.

* In the related work, it is claimed that Bayesian approaches to loss
minimization are not applicable because sampling is harder
computationally than MAP. But isn't the whole point of random MAP
models that we can do efficient sampling? It actually seems that the
most natural way of optimizing non-decomposable losses within the
random MAP framework would be to learn parameters using the previous
random MAP learning, then draw many samples for a test example and use
the Bayesian approach on these samples (choosing the sample that
minimizes expected loss under the set of drawn samples). This would
be a straightforward-to-implement baseline that I think should be
added.
Summary: There are several contributions here, which together provide a set of tools for optimizing expected loss under random MAP perturbation models. None of the components is particularly ground-breaking, but they appear natural and correct, and I think many people will find it interesting.
Author Feedback

Author rebuttal: We thank the reviewers for the time and effort invested in reviewing our submission. We agree with all reviewers on the merit of the paper. Below we address the different concerns that were raised by the reviewers:

Assigned_Reviewer_6
1) We assume that the reviewer meant a natural comparison to be [16] (Perturb-and-MAP, Papandreou & Yuille) rather than [6] (Fixing max-product, Globerson & Jaakkola). We apologize if we misread these remarks.
2) [16] utilized random MAP perturbations to moment matching and [22] to likelihood. None of these works dealt with loss minimization, and our work is a natural continuation of this direction to loss minimization. In retrospect we should have compared to these works, but as Assigned_Reviewer_7 pointed out - "using the “right” loss function has been demonstrated to be important by other researchers."
3) Thank you for pointing out the clarity problem with regard to corollary 1, we will change the statement "every smooth posterior distribution" to make it clear that we only refer to the additive posteriors that are described in corollary 1. A similar statement in the related work section refers to Theorem 2. We will clarify that as well.
4) Although we pointed it out in the discussion, we will make it clearer that no solid example for 4.1 was described. Our work focus on random MAP perturbation posterior models and it is natural to ask how it can be applied when MAP can only be approximated. Although we do not provide a solid example, we believe that further research in this direction should be encouraged.
5) Theorem 3 does not stem from other theorems in our submission, thus to avoid unnecessary references we did not state it as a corollary of a previous work. We kindly note that this result is of interest to the approximate inference NIPS community, and extends one of the results in Globerson & Jaakkola [6]. Thank you for suggesting to rearrange the sections, we are working on it.
6) We agree that our work extends [8] beyond Gaussian additive posterior models. This is an important extension: for example, Gaussian posterior models cannot be easily applied to sub-modular settings which are of considerable interest in the NIPS community.

Assigned_Reviewer_7
We do not extend PAC-Bayesian theory to tighter bounds. However, we extend the applicability of the PAC-Bayesian approach to different posterior models. Specifically, we suggest to use this approach to improve the learning of segmentations, a task for which previous PAC-Bayesian theory could not have been applied efficiently without our suggested multiplicative posteriors to the pairwise weights.

Thank you for pointing out the lack of definition for p(gamma) in Theorem 1.

Assigned_Reviewer_8
Thank you for pointing out that dealing with approximate inference and positivity constraints are important practical concerns, and the methods presented for doing so are natural.
1) We will add a more elaborate discussion on differences of the PAC-Bayesian approaches (i.e., the Catoni bound we are using and the McAllester bound that is used in [8]).
2) The variance of stochastic descent methods turns to be a central question. It was not a major problem in our segmentation experiment but we did encounter this problem in another setting where w is of size of millions, and we are investigating few directions for this specific problem (we hope that we will be able to use ideas from Paisley 2012 for this). We agree that such a large-scale experiment would reveal interesting things about the learning.
3) We are happy to try the straight forward baseline.
4) We agree with your detailed comments, thanks. We will add the citations.